# Speeding up adiabatic ion transport in macroscopic multi-Penning-trap stacks for high-precision experiments
Moritz von Boehn [1] ✉, Jan Schaper [1], Julia A. Coenders[1], Johannes Brombacher [1], Teresa Meiners[1], Malte Niemann[1], Juan M. Cornejo [1] ✉, Stefan Ulmer[2,3] & Christian Ospelkaus [1,4] ✉

Multi-Penning traps are an excellent tool for high-precision tests of fundamental physics in a variety of applications, ranging from atomic mass measurements to symmetry tests. In such experiments, single ions are transferred between distinct trap regions as part of the experimental sequence, resulting in measurement dead time and heating of the ion motions. Here, we report a procedure to reduce the duration of adiabatic single-ion transport in macroscopic multi-Penning-trap stacks by using ion-transport waveforms and electronic filter predistortion. For this purpose, transport adiabaticity of a single laser-cooled $^9Be^+$ is analyzed via Doppler-broadened sideband spectra obtained by stimulated Raman spectroscopy, yielding an average heating per transport of $2.6 \pm 4.0$ quanta for transport times between 7 and 15 ms. Applying these techniques to current multi-Penning trap experiments could reduce ion transport times by up to three orders of magnitude. Furthermore, these results are a key requisite for implementing quantum logic spectroscopy in Penning trap experiments.

Penning trap based experiments have enabled ultrahigh precision measurements on atomic ions with applications in nuclear and particle physics[1,2], as well as fundamental constants[3–5] and symmetry tests in the framework of standard model physics[6,7]. These experiments rely on precise measurements of the trapped particles' motional frequencies by using high-sensitivity superconducting resonators in Penning traps with different interconnected trap regions, where each region fulfills a specific task of the experimental procedure[8–12]. Here, the transport between these regions is a crucial part in the experimental scheme, since it results in heating of the ion eigenmotions and, due to the transport duration, to a decreased duty cycle. To circumvent either, depending on the experimental requirements, two approaches can be applied. One where the ion is shuttled quickly within microseconds by fast trap voltage switching[13,14], and the other where the transport is designed such that the induced heating is reduced to a minimum by slow voltage ramps[15]. The former comes with the disadvantage of motional heating on the order of tens of meV[14], while for the latter, each transport can take up to several tens of seconds or even minutes[6,15].

In ion-based quantum computing, the development of ion transport techniques has allowed to combine short transport times and low transport-induced heating in micro-fabricated electrode traps[16–22]. Applying these fast and adiabatic transport methods to high-precision multi-Penning-trap experiments would allow to further increase the measurement duty cycle,

with potential application in atomic mass measurements[1,2], as well as g-factor and charge-to-mass ratio measurements of particles and antiparticles for CPT (Charge-Parity-Time) invariance tests[6,7,23]. However, such experiments are typically conducted in macroscopic cylindrical traps with a substantially different electrode architecture[22], where ions need to be shuttled over larger distances between the trap regions. Additionally, these experiments also rely on optimal conditions regarding confinement and isolation, which are achieved by trap-electrode multi-stage RC low-pass filters with low cut-off frequencies of a few Hz[24]. A drawback of the low cut-off is a resulting high rise time limiting the achievable transport duration, as fast changing trap electrode voltages waveforms, necessary to implement the ion transport, become distorted and cause heating of the ion motion. This can be circumvented by predistorting the transport waveforms in order to counteract the low-pass filtering behavior[18,25]. The aforementioned approach will soon become even more relevant with recent advances in laser-based ion cooling in ultrahigh precision Penning trap experiments[26–28]. Such techniques enable cooling times in the millisecond regime, making fast transport a crucial factor in further increasing measurement sampling rates. In addition, fast and adiabatic transport methods are required to engineer full motional control methods in Penning traps to improve particle localization and to utilize quantum logic spectroscopy[29–31]. The latter is promising to boost current g-factor measurements with (anti-)protons for discrete

[1]Institut für Quantenoptik, Leibniz Universität Hannover, Hannover, Germany. [2]RIKEN, Ulmer Fundamental Symmetries Laboratory, Wako, Japan. [3]Institut für Experimentalphysik, Heinrich Heine Universität Düsseldorf, Düsseldorf, Germany. [4]Physikalisch-Technische Bundesanstalt, Braunschweig, Germany. ✉e-mail: vonboehn@iqo.uni-hannover.de; cornejo-garcia@iqo.uni-hannover.de; christian.ospelkaus@iqo.uni-hannover.de

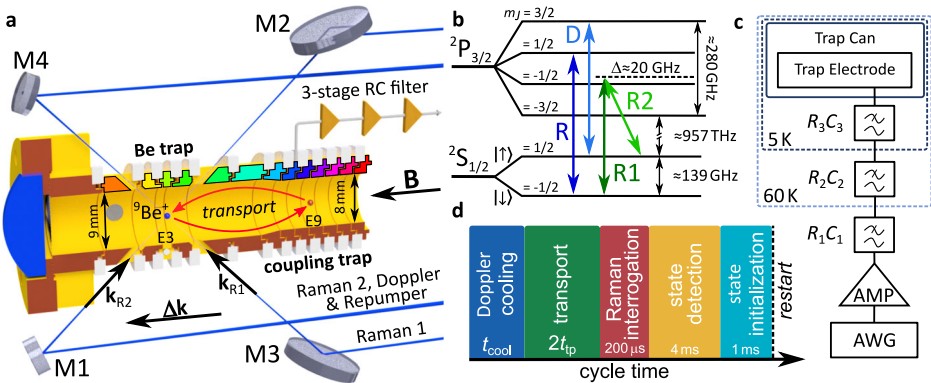

**Fig. 1 | Experimental setup, energy levels and experimental sequence. a** Cross-section view of the Be and coupling traps. Electrodes are cylinder-shaped and made of gold-plated oxygen-free high thermal conductivity copper. They have an inner diameter of 9 mm for the Be trap and 8 mm for the coupling trap. The single laser-cooled $^9$Be$^+$ion is transported back and forth over a distance of 22.3 mm from electrode 3 (E3) in the Be trap to electrode 9 (E9) in the coupling trap. Each electrode is equipped with a 3-stage low-pass RC filter to minimize electrical noise. The trap electrodes are color-coded for identification. **b** Relevant internal level scheme of $^9$Be$^+$in a 5 Tesla magnetic field. The cooling and detection (D) as well as the repumper transition (R) are represented by light and dark blue arrows, respectively. The Raman transitions are depicted by dark and light green arrows for Raman transitions 1 (R1) and 2 (R2), respectively. Energy levels are not to scale. **c** Simplified schematic of the trap voltage electronics. The electrode voltages, generated by an arbitrary waveform generator (AWG), are amplified and applied to the electrodes at cryogenic temperatures using filters at different temperature stages (room temperature, 60 K and 5 K). $R_1$, $R_2$ and $R_3$ are 5.0 kΩ, 5.2 kΩ and 7.1 kΩ, respectively. $C_1$, $C_2$ and $C_3$ are 4.8 nF. **d** Measurement cycle scheme, where $t_{\rm cool}$ and $t_{\rm tp}$ are the cooling and transport times, respectively.

symmetry tests to the quantum limit by using single laser-cooled $^9$Be$^+$ions for sympathetic cooling schemes and quantum-logic state detection[31,32].

   In this letter, we apply fast and adiabatic transport techniques to a single laser-cooled $^9$Be$^+$ion in a macroscopic cryogenic multi-Penning-trap stack and verify adiabaticity of the transport by analyzing Doppler-broadened resolved motional sideband spectra to measure the axial temperature of the ion. An average heating per transport of 2.0 ± 3.2 quanta is achieved for an ion transport between two regions of our multi-Penning-trap stack separated by 2.2 cm and durations between 10 and 15 ms. Applying multistage RC-low-pass filter predistortion to the transport voltage waveforms, the transport duration can be lowered to 7 ms, yielding an average heating per transport of 2.6 ± 4.0 quanta.

## Results and discussion
### Working principles and experimental setup
Penning traps use a combination of a static electric quadrupole field and a magnetic field **B** of field strength $B$ of up to several Tesla to trap charged particles. The combination of both fields creates a 3D confining potential in which the particles undergo a motion which can be decomposed in three harmonic motions: The axial motion along the direction of **B** with characteristic frequency $v_z$ and two radial motions, perpendicular to **B**, called magnetron and reduced cyclotron motion with characteristic frequencies $v_-$ and $v_+$, respectively. Using the invariance theorem, all three motional frequencies determine the free cyclotron frequency $v_c = \sqrt{v_z^2 + v_-^2 + v_+^2} = qB/(2\pi m)$[33], which depends solely on the particle's charge-to-mass ratio $q/m$ and the magnetic field strength.

   Figure 1a shows a cut section view of the so-called "Be trap" with laser access as well as the neighboring so-called "coupling trap" to which the ion is shuttled (red arrows in Fig. 1a). More details about the experimental setup can be found elsewhere[34,35]. The relevant energy levels and transitions of a $^9$Be$^+$ion at a magnetic field strength of $B = 5$ T are depicted in Fig. 1b. At first, an ion is loaded by ablation into the "Be trap". Afterwards, 3D Doppler cooling is performed using the Doppler and repumper laser beams which share a common beam path via mirrors M1 and M2 (see Fig. 1a), resulting in an angle of 45° relative to the trap axis. For sufficient cooling of the magnetron motion, a vertical offset to the trap center is used[35]. A resolved motional sideband spectrum is obtained by coupling the particle's motion to its internal degree of freedom. This can be done by a two-photon stimulated Raman process[29]. Here, the interaction of the trapped particles with two laser fields $\mathbf{E}_{1,2}$ far detuned from an auxiliary level by $\Delta \approx 20$ GHz results in coupling of two electronic states $|\uparrow\rangle$ and $|\downarrow\rangle$, and motional states $|n_{1,2}\rangle$.

Apart from the resonance condition of the Raman transition $(v_1 - v_2) - v_0 = v_z(n_1 - n_2)$, where $v_1$ and $v_2$ are the Raman laser frequencies and $v_0$ is the frequency of the carrier transition $|\downarrow\rangle \leftrightarrow |\uparrow\rangle$, both laser fields need to have a finite projection of their wave vector difference $\Delta\mathbf{k} = \mathbf{k}_{R1} - \mathbf{k}_{R2}$ onto the motional mode of interest[35–37]. To accomplish this, the Raman 2 (R2) laser beam follows the beam path of the Doppler and repumper beams and the Raman 1 (R1) laser beam follows a path via mirrors M3 and M4 in order to cross R2 under an angle of 90°, resulting in the desired wave vector difference $\Delta\mathbf{k}\|\mathbf{B}$, necessary to address the axial motion. In case of a Doppler laser-cooled particle, the sideband spectrum follows a Doppler-broadened envelope whose width is determined by the particle's mode temperature[36,38]. More details can be found elsewhere[35,36].

   The transport of individual trapped particles through a Penning trap stack has been discussed previously[39]. To summarize, the goal is to produce a moving harmonic trapping potential along the transport path where the charged particle is confined. In the case of a cylindrical Penning trap, this is a one-dimensional problem, since the only way to perform the transport adiabatically is to move the particle parallel to the magnetic field lines, while the radial confinement is achieved by the magnetic field. For this, the voltage of each trap electrode must be calculated for a harmonic confinement of the particle at different points along the axial direction. As a result, a voltage waveform is calculated for each electrode, where the transport time defines the duration of the voltage waveform. For the case of a multi-stage RC low-pass filter with $n$ stages, the calculated waveform voltage $V_{\rm out}(t)$ for each electrode must be modified to $V_{\rm in}(t)$ given by

$$V_{\rm in}(t) = \left[\prod_{i=1}^{n}\left(1 + R_i C_i \nabla_t\right)\right] V_{\rm out}(t), \tag{1}$$

which is the input voltage waveform to the RC filters, where $R_i$ and $C_i$ are the resistance and capacitance of the $i$-th filter stage and $\nabla_t^i V_{\rm out} \equiv d^i V_{\rm out}/dt^i$. The cut-off frequency of each filter stage is given by $f_c^i = 1/(2\pi R_i C_i)$, which is related to the rise time $\tau_i = 1/(2\pi f_c^i)$. For the Penning trap utilized in this work, three filter stages are used as shown schematically in Fig. 1c. Note that there are experiments using up to four stages[15]. Figure 2a shows the voltage waveform applied to each trap electrode to transport a $^9$Be$^+$ion between the Be and coupling traps in 7 ms at an axial trap frequency of 435 kHz with filter predistortion using Eq. (1). The difference between the voltage waveforms with and without filter predistortion (dashed lines in Fig. 2a) is shown in Fig. 2b. Here, it can be seen that for short transport times it is

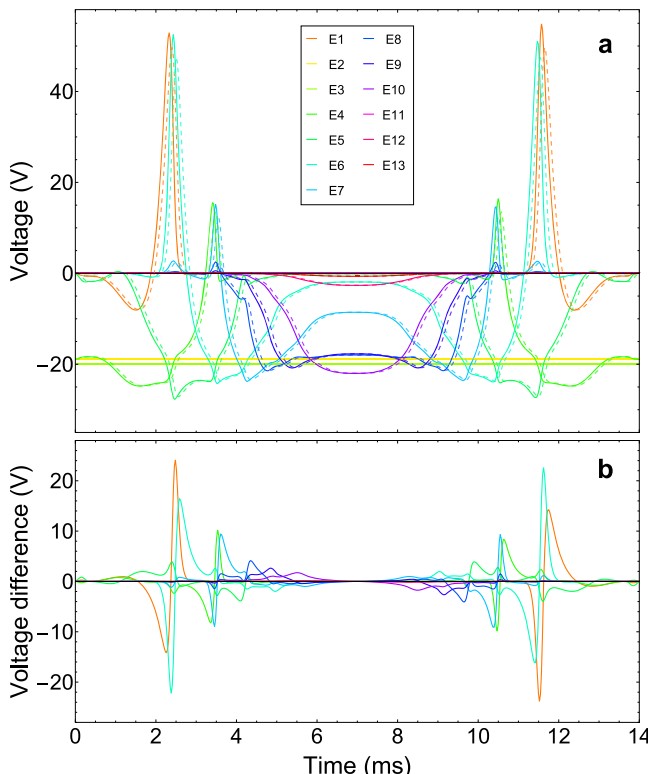

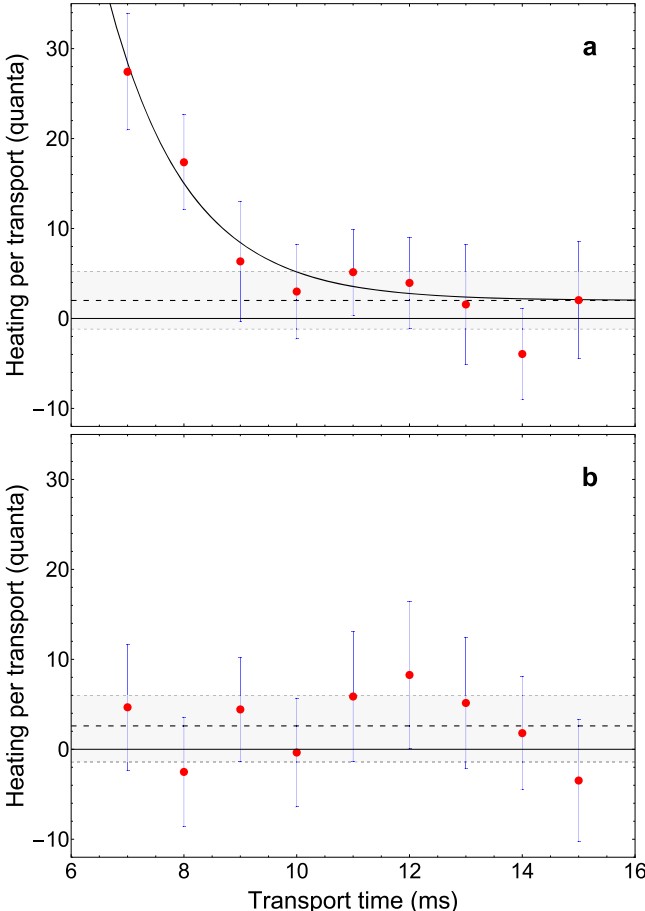

**Fig. 2 | Voltage waveforms. a** Voltage waveforms with (solid lines) and without (dashed lines) filter predistortion as a function of time to transport a $^9Be^+$ion back and forth between the Be and coupling traps within $2t_{tp} = 14$ ms at an axial trap frequency of 435 kHz. **b** Voltage difference between voltage waveforms with and without filter predistortion for the same transport time and trap frequency. Different colors denote different electrodes according to the color code shown in Fig. 1a.

**Fig. 3 | Heating per transport. a** Heating per transport for several transport times $t_{tp}$ using voltage waveforms without filter predistortion. Each data point (red, representing the mean) and the respective error (blue, ± standard deviation) are obtained from the Gaussian envelope of a sideband spectrum acquired after ion transport. A total of around 20 sidebands per spectrum were probed, where each sideband scan is obtained from 1250 Raman interrogations. The black dashed line shows the transport-induced heating averaged for 10 ms $\leq t_{tp} \leq 15$ ms transport times. Gray dashed lines filled in light gray show the error bands of the averages with a confidence level of ± one standard deviation. The solid black line shows an exponential decay fit to the data to guide the eye. **b** Heating per transport for several transport times $t_{tp}$ using voltage waveforms with filter predistortion. The black dashed line shows the transport-induced heating averaged for all transport times.

necessary to use voltage amplitude differences of up to several tens of volts in order to balance the delaying effect of the filters. For this, a 3-stage RC low-pass filter with a cut-off frequency of around 1.2 kHz shown schematically in Fig. 1c was used for the waveform calculations. $R_2$ and $R_3$ were estimated from measured resistance values for several temperatures close to 60 K and 5 K, respectively.

**Measurement of transport induced heating**

The ion heating induced by the transport is determined by comparing the ion's axial temperature after transport to a reference temperature $T_{ref}$ without transport. The temperature measurements are done by analyzing Doppler-broadened sideband spectra acquired by sideband spectroscopy[35,36]. Further details can be found in Section Methods. $T_{ref}$ was found to be $1.7 \pm 0.1$ mK, which corresponds to the Doppler cooling temperature which is around 3 times larger than the expected Doppler cooling limit of 0.5 mK. The difference between the observed temperature and the theoretical limit arises from the complicated radial cooling dynamics of a single $^9Be^+$ion in Penning traps[35,36]. Sideband spectra after transport are acquired using a measurement cycle shown in Fig. 1d. Each interrogation cycle starts with Doppler cooling with a duration $t_{cool}$ of a few milliseconds to initialize the temperature of the ion. This phase is followed by an ion transport between electrodes E3 and E9, and back to electrode E3 (see Fig. 1a) for a variable transport time $2t_{tp}$, where $t_{tp}$ is the duration for each transport (forth: E3 $\rightarrow$ E9 and back: E9 $\rightarrow$ E3). Afterwards, the Raman transition is probed for an interaction time of 200 μs. For this, both Raman laser beams R1 and R2 are focused to a waist of $\approx 150$ μm at the position of the ion and stabilized to powers of 3 mW and 1 mW, respectively. The difference in power needs to be set to fulfill polarization requirements of each transition[36]. To read out the ion's spin state after the interrogation, it is exposed to the Doppler laser for 4 ms. As $|\downarrow\rangle$ is a dark state for the cooling

light, detecting no fluorescence on a photomultiplier tube refers to the ion being in $|\downarrow\rangle$ and $|\uparrow\rangle$ otherwise. Finally, the ion is initialized in $|\uparrow\rangle$ by a 1 ms repumping pulse, which closes the cycle. The averaged binary results of the detection phase of each cycle allow to determine the excitation probability of the sideband transition. As for shorter transport times heating and thus an increase in Doppler broadening is observed, additional higher-order sidebands become visible and the total number of scanned transitions needs to be increased depending on $t_{tp}$. Figure 3 shows the heating per transport $\Delta T = (T_{tp} - T_{ref})/2$, which is the difference of the evaluated axial mode temperatures after transport $T_{tp}$ and the reference temperature, divided by the number of transports. A factor of 2 is considered since $T_{tp}$ is measured after shuttling from electrodes E3 to E9 and back to electrode E3 in a duration of $2t_{tp}$. Here, the relation $k_B\Delta T = \hbar\omega_z$ is used to show the data in units of motional quanta, where $\hbar$ is the reduced Planck constant, $k_B$ is the Boltzmann constant and $\omega_z = 2\pi\nu_z$. The measured heating rate of the trap of $\approx 5$ quanta per second[37] has been neglected. Without filter predistortion (Fig. 3a), for $t_{tp} \leq 9$ ms, a rapid increase of the induced heating can be observed as a consequence of the distortion of the applied voltage waveforms by the 3-stage low-pass filter. Using $t_{tp} \leq 6$ ms results in a drastic

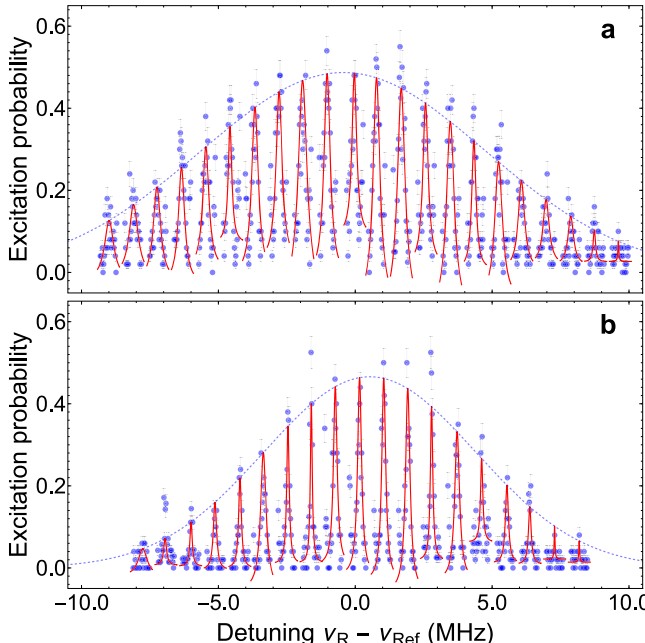

**Fig. 4 | Resolved sideband spectra. a** Resolved sideband spectrum after transport for a transport time of $t_{tp} = 7$ ms without filter predistortion. **b** Resolved sideband spectrum after transport for a transport time of $t_{tp} = 15$ ms using filter predistortion. Each data point (blue, representing the mean) and the respective error (clear blue, ± standard deviation of the mean) are obtained from 50 Raman interrogations after transport. Red lines are Lorentzian fits to the sideband data, clear blue dashed lines are Gaussian envelope fits to the sideband maxima. $\nu_{Ref} = 138.909014$ GHz is a reference frequency close to $\nu_0$.

increase of heating such that the ion's motion cannot be recooled within $t_{cool}$ and no sideband resonances could be observed. An average heating of $2.0 \pm 3.2$ quanta for transport times between $10$ ms $\leq t_{tp} \leq 15$ ms is obtained. In contrast, using waveforms with filter predistortion (Fig. 3b), no rapid increase of the induced heating can be observed and an average heating of $2.6 \pm 4.0$ quanta is obtained for transport times $7$ ms $\leq t_{tp} \leq 15$ ms. Using $t_{tp} \leq 6$ ms also results in a drastic increase of heating, but in this case, heating can be measured for $t_{tp} = 6$ ms to be $160 \pm 20$ quanta (not shown in Fig. 3). The rapid increase in heating for $t_{tp} \leq 6$ ms can be attributed to a severe distortion of the waveforms for shorter transport times due to the filter cut-off frequency, such that the distortion is too large to be fully compensated. Nevertheless, these results demonstrate that the application of filter pre-distortion to the transport waveforms reduces transport times by at least 30 % while maintaining comparable levels of average heating. In addition, it is important to note that the observed heating of a few quanta is comparable to the uncertainty of our analysis method (see Section Methods), where each ion temperature measurement is an individual and independent measurement, with the majority as well as the respective error being comparable to the reference temperature.

## Discussion

We have demonstrated faster adiabatic transport in the quanta regime of a single ion in a macroscopic cryogenic multi-Penning-trap stack by applying ion-based quantum computing and filter predistortion techniques. The implementation of these techniques in high-precision multi-Penning-trap experiments will allow to reduce the measurement dead time and enhance the sampling rate with negligible heating. In such experiments, electrical noise is suppressed by multi-stage low-pass filters for the trap electrode biasing lines with a low cut-off frequencies of a few Hz[24]. Based on our findings presented in this work and taking into account the relation between the cut-off frequencies of our experimental setup $f_c^h \approx 1.2$ kHz and an experimental setup with a low cut-off frequency $f_c^l \approx 0.01 f_c^h$ for optimal measurement

conditions, ion transport times on the order of $\approx 700$ ms could be achieved without significant changes to the electrode biasing lines. This is between one and two orders of magnitude lower than transport times achieved in state of the art experiments[6,15,40]. Moreover, using these transport techniques combined with fast switching between electrode biasing lines with RC low-pass filters at low ($f_c^l$) and high ($\gtrsim f_c^h$) cut-off frequencies will allow to further decrease the duration of ion transport between different trap regions to a few milliseconds with negligible heating. This is three orders of magnitude lower than the lowest transport time achieved in current high-precision experiments[40]. In addition, the presented results are compatible with transport of a single ion in its motional ground state in the millisecond regime, which is a key requisite for the development of quantum-logic inspired techniques in Penning traps, where full motional control at the single quantum level of atomic ions in different Penning trap areas is required[31].

## Methods

Each measurement of the axial ion temperature $T_z$ is performed by analyzing the Doppler-broadened Gaussian envelope of a motional sideband spectrum. For this, sidebands are probed until excitation cannot be observed anymore. It is important to note that in order to reduce the total measurement time, and therefore related errors arising from, e.g., laser beam pointing fluctuations, only every second sideband is probed. This can be accomplished because the sideband transitions are separated in frequency by integer multiples of the axial frequency. In total, for each spectrum around 20 sidebands are probed, where the number of sideband transitions depends on the Doppler broadening and thus the ion's axial temperature. For comparison, Fig. 4 shows the resolved sideband spectrum measured after transport with the maximum and minimum heating observed in Fig. 3. The relative shift of the Gaussian fit center frequency in Fig. 4a and b results from slow magnetic field drifts, since both measurements were performed several weeks apart. In order to calculate the axial ion temperature $T_z$, the relation $T_z = m\lambda^2\nu_D^2/(8\,2\,k_B)$ is used[36], where $m$ is the mass of the $^9\text{Be}^+$ ion, $\lambda \approx 313$ nm is the transition wavelength $^2S_{1/2} \leftrightarrow ^2P_{3/2}$ (see Fig. 1b) and $\nu_D$ is the full width at half maximum of the fitted Gaussian envelope. For Fig. 4a, b, the values of $\nu_D$ are determined to be $(12.3 \pm 0.6)$ MHz and $(9.1 \pm 0.5)$ MHz, respectively, from which the corresponding ion temperatures $T_z$ of $(2.88 \pm 0.19)$ mK and $(1.59 \pm 0.18)$ mK are calculated. Together with $T_{ref}$, the heating resulting from each transport is calculated and shown in Fig. 3.

## Data availability

The data that support the findings of this study are available upon reasonable request from the authors.

## Code availability

The code that support the findings of this study are available upon reasonable request from the authors.

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

## Acknowledgements

We are grateful for discussions with J.J. Bollinger, R.C. Thompson and P.O. Schmidt. This work was supported by PTB, LUH, and DFG through the clusters of excellence QUEST and QuantumFrontiers as well as through the Collaborative Research Center SFB1227 (DQ-mat Project-ID 274200144) and ERC StG "QLEDS". We also acknowledge financial support from the RIKEN Pioneering Project Funding and the MPG/RIKEN/PTB Center for Time, Constants and Fundamental Symmetries.

## Author contributions

M.B. and J.S. have experimentally implemented ion transport in our setup. J.M.C. has calculated the waveforms for ion transport. M.B., J.S., J.A.C., and J.M.C. have implemented and characterized the laser system. S.U., M.N., T.M., J.B., and J.M.C. have worked on the construction of the Penning trap stack. S.U., J.M.C., and C.O. have guided the project. C.O. initiated and supervised the project. All authors have discussed the findings of the manuscript.

## Funding

## Competing interests

The authors declare no competing interests.
