## [Transparent Peer Review file · Communications Physics]

Speeding up adiabatic ion transport in macroscopic multi-Penning-trap stacks for high-precision experiments

Corresponding Author: Mr Moritz von Boehn

Version 0:

Reviewer comments:

Reviewer #1

(Remarks to the Author)

The authors demonstrate a protocol for fast transport of a single Be⁺ in a Penning trap by designing waveforms facilitating fast transport and RC filtering to predistort the waveforms which are further distorted by the filters in the trap isolating it from the environment. The authors report a heating of ~2-3 quanta during the transport. The manuscript is well written and easy to follow. The data presented and the analysis is sound. In short, the paper meets the criteria for publication in my opinion, however I would like the authors to comment on several points.

1. Can the authors comment on the merits of their procedure as compared to what has been achieved in for example Ref. 22 in the context of Penning traps for quantum computation? Is it correct to say that while the observed heating here is larger (in absolute number of quanta) that the distance the ion is moved is larger and it was performed faster?
2. Can the authors comment on the typical heating rate of the trap? How does the heating due to the shuttling compare to the heating rate?
3. Is this additional heating a hindrance for the precision measurement experiments? If so, what are the additional cooling protocols that need to be performed and what are the time scales? In short, if the ion needs to be cooled to near its motional ground state via a sub Doppler cooling scheme which takes a few ms, is the faster transport still the optimal method?

Reviewer #2

(Remarks to the Author)

Here's a revised version of your text:

The manuscript "Speeding Up Adiabatic Ion Transport in Macroscopic Multi-Penning-Trap Stacks for High-Precision Experiments," submitted by Moritz et al., presents an ion transport experiment characterized by a low heating rate. In this work, the authors verify that the filter predistortion method for control voltage is effective. This technique will be quite useful for further research in atomic mass measurements, quantum information processing with Penning traps, and other areas of fundamental physics. I believe this progress is well-suited for Communications Physics.

The manuscript is well prepared; however, I still have some questions regarding their experiment to measure the temperature of the ions. The authors mention, "The temperature measurements are done by analyzing Doppler broadened sideband spectra acquired by sideband spectroscopy." Is the same method used as in their previous study referenced in Ref. 36? Additionally, they state, "Probing only every second sideband comes with the advantage of a reduced total measurement time." Does this mean that after obtaining the entire spectrum with 20 transitions, they only scanned every second sideband in subsequent measurements to determine temperatures for each data point?

Regarding the sentence after Eq. 1, I believe there should be no white space before "which is the input..." as it should be part of the same paragraph as the displayed equation.

Reviewer #3

(Remarks to the Author)

The authors present a novel technical achievement in the context of a precision measurement experiments, where a Be⁺ ion

is transported within two separate stages of a multi-Penning trap stack. This is necessary as, in this and similar experiments, ions of different species are produced and held in different sections of the trap, to be then brought together to perform the measurement.

This work improves on results already presented by the same group, described in ref. [39]. The novelty in the current paper stands in the author's achievement of faster ion transport, still within the adiabatic regime, due to the inclusion of filter precompensation. The technique itself is not new, as it has been already demonstrated in the context of ion transport experiment focused on quantum computing (see e.g. ref [18] or <https://doi.org/10.1103/PhysRevA.102.022611>), where the transport distances and timescales are quite different. Nonetheless, their contribution is original and of high interest in their community, as (at the best of my knowledge) this is the first time such techniques are applied to trapped-ion-based precision measurement experiments, with demonstrated favorable results.

Their methodology is valid and clear, based on known techniques for the measurement of motional excitations in trapped ions, and their results are convincing. Overall, I would recommend the paper for publication. However, before the final publication, I suggest improving on a few points that in my opinion would help to increase the quality of the manuscript.

1. I think the paper lacks details on the temperature measurements. It is true that the technique is referenced elsewhere, e.g. in ref. [35], but it would be beneficial for this publication to give more in-context details, especially in view of directing the paper to a non-specialized audience. For example, the authors could show the data of one Doppler-broadened sideband spectrum as a representative measurement.

2. Can the authors comment on the drastic increase in heating seen for $t_{tp} \leq 6$ ms, regardless of precompensation? Can they relate this timescale with other waveform parameters, e.g. the filter cutoff frequency?

3. In Fig.2, it would be clearer to plot on the same axes the voltage waveforms with and without filter precompensation, rather than plotting the difference between the two on a separate axis. Furthermore, it would be useful to add a color-coded sketch of the trap electrodes, perhaps similar to the one in ref. [39], fig 1, matching the electrode colors with the corresponding waveform plot – or with a similar visual reference.

Version 1:

Reviewer comments:

Reviewer #1

(Remarks to the Author)

The authors have addressed all my questions and comments.

Reviewer #2

(Remarks to the Author)

The authors have addressed all of my concerns in their responses, and the new version of the manuscript is clarified. Therefore, I recommend the manuscript for publication.

Reviewer #3

(Remarks to the Author)

I thank the authors for the time taken and the effort put in replying to all referees' comments. I think that this clarified all questioned points and definitely improved the manuscript. I recommend their work for publication.

Dear editors of Communications Physics,

We would like to thank you for your valuable time spent reviewing our manuscript. Below we included a point-by-point response to the referee comments. Our comments and changes to the manuscript are color coded. In addition, we have divided the manuscript into sections following the style guidelines of the journal. With best regards from the authors,

Moritz von Boehn,
Juan M. Cornejo and
Christian Ospelkaus

Reviewer #1

The authors demonstrate a protocol for fast transport of a single Be^+ in a Penning trap by designing waveforms facilitating fast transport and RC filtering to predistort the waveforms which are further distorted by the filters in the trap isolating it from the environment. The authors report a heating of ~ 2 -3 quanta during the transport. The manuscript is well written and easy to follow. The data presented and the analysis is sound. In short, the paper meets the criteria for publication in my opinion, however I would like the authors to comment on several points.

We thank the reviewer for the careful review of our manuscript and for the positive comments. We hope that the revised manuscript will be suitable. Below are our responses to each of the reviewer's comments.

Comments on the manuscript follow.

1. Can the authors comment on the merits of their procedure as compared to what has been achieved in for example Ref. 22 in the context of Penning traps for quantum computation? Is it correct to say that while the observed heating here is larger (in absolute number of quanta) that the distance the ion is moved is larger and it was performed faster?

Response: With respect to Ref. 22, our transport was done around 30 times faster while the transport distance was of around 73 times larger. However, the surface trap of Ref. 22 and our macroscopic Penning trap are wildly different in scale and architecture. Therefore, we consider a direct comparison to be of limited value due to the significant differences in trap geometry and size. For this reason, we included Ref. 22 regarding the progress toward the development of ion-based quantum computing. With respect to the amount of heating, note that our data is compatible with vanishing heating over the useful transport time range. Please also see reply to comment 3. To provide further clarification, the following text has been included in the manuscript.

Previous text (Page 1, line 49): “In ion-based quantum computing, the development of ion transport techniques have allowed to combine short transport times and low transport-induced heating [16–22]. Applying these fast... invariance tests [6, 7, 23]. However, such experiments rely on optimal conditions ...”

New text (Page 1, line 51): “In ion-based quantum computing, the development of ion transport techniques has allowed to combine short transport times and low transport-induced heating in micro-fabricated electrode traps [16–22]. Applying these fast ... invariance tests [6, 7, 23].”

However, such experiments are typically conducted in macroscopic cylindrical traps with a substantially different electrode architecture (see e.g. Ref. 22), where ions need to be shuttled over larger distances between the trap regions. Additionally, these experiments also rely on optimal conditions...

2. Can the authors comment on the typical heating rate of the trap? How does the heating due to the shuttling compare to the heating rate?

Response: The heating rate in our trap was measured to be (5.0 ± 0.3) quanta per second in Ref. 37. For the transport times applied in this work, heating is expected to be around 0.075 quanta for a transport time of 15 ms due to the heating rate of the trap. In order to clarify this, we introduced the following text in the manuscript.

Previous text (Page 4, line 8): "...the Boltzmann constant and $\omega_z = 2\pi\nu_z$. Without filter predistortion (Fig. 3a), ..."

New text (Page 4, line 18): "...the Boltzmann constant and $\omega_z = 2\pi\nu_z$. The measured heating rate of the trap of around ≈ 5 quanta/s [37] has been neglected. Without filter predistortion (Fig. 3a), ..."

3. Is this additional heating a hinderance for the precision measurement experiments? If so, what are the additional cooling protocols that need to be performed and what are the time scales? In short, if the ion needs to be cooled to near its motional ground state via a sub Doppler cooling scheme which takes a few ms, is the faster transport still the optimal method?

Response: For the implementation of the transport protocols discussed in this work in high-precision experiments with Penning traps in general, it can be reasonably assumed that the heating of a few quanta per second in the axial motion per transport can be neglected. However, the reduction in transport time by applying the transport protocols discussed in this work will allow to significantly reduce measurement dead times, which are of increasing importance in current experiments with the development of laser-based ion cooling techniques (see e.g. Ref. 26). We have modified the text accordingly in order to make the point more clear.

Previous text: (Page 4, line 42): "... dead time and enhance the sampling rate. In such experiments, ..."

New text: (Page 5, line 3): "... dead time and enhance the sampling rate with negligible heating. In such experiments, ..."

Response: The use of fast transport protocols discussed in this work is mandatory for the application of quantum logic cooling and detection techniques in high precision experiments with Penning traps. The ion transport must be performed in the millisecond regime to reduce the dead time of the whole measurement (see e.g. ref. 31). But also, in these measurements the ion has to be cooled close to its motional ground state. The results of this work demonstrate that both requirements are feasible in a macroscopic Penning trap stack. For this, it is important to note that the observed average heating is comparable to the error of our analysis method of a few quanta. In addition, each ion transport heating measurement represents an individual and independent measurement and, for the majority, the measured temperatures and their respective errors are comparable to the reference temperature. Hence, we believe that the presented

method is applicable to be used together with sub Doppler cooling schemes in macroscopic Penning trap stacks. We introduced the following text in the manuscript.

Previous text (Page 4, line 29): "... of average heating."

New text (Page 4, line 42): "of average heating. In addition, it is important to note that the observed heating of a few quanta is comparable to the uncertainty of our analysis method (see Section Methods), where each ion temperature measurement is an individual and independent measurement, with the majority as well as the respective error being comparable to the reference temperature.

Reviewer #2

The manuscript "Speeding Up Adiabatic Ion Transport in Macroscopic Multi-Penning-Trap Stacks for High-Precision Experiments," submitted by Moritz et al., presents an ion transport experiment characterized by a low heating rate. In this work, the authors verify that the filter predistortion method for control voltage is effective. This technique will be quite useful for further research in atomic mass measurements, quantum information processing with Penning traps, and other areas of fundamental physics. I believe this progress is well-suited for Communications Physics.

The manuscript is well prepared; however, I still have some questions regarding their experiment to measure the temperature of the ions.

We are grateful to the reviewer for taking the time to carefully review our manuscript and for his/her positive feedback. We hope that the revised manuscript will meet his/her expectations. Please find our responses to each of the reviewer's comments below.

Comments on the manuscript follow.

1. *The authors mention, "The temperature measurements are done by analyzing Doppler broadened sideband spectra acquired by sideband spectroscopy." Is the same method used as in their previous study referenced in Ref. 36?*

Response: We used the same method as in Ref.36. We have added the references in this part of the manuscript and also a new method section.

Previous text (Page 3, line 25): "The temperature measurements are done by analyzing Doppler broadened sideband spectra acquired by sideband spectroscopy. For this, the carrier transition as well as every second sideband are probed until no excitation can be observed anymore. Here, around 20 transitions (including the carrier transition) are scanned. Probing only every second sideband comes with the advantage of a reduced total measurement time. Laser beam pointing fluctuations of a few tens of μm within an hour are observed, resulting in fluctuations of the observed excitation probability due to a decreased overlap of both Raman beams and the ion. T_{ref} was found to be ..."

New text (Page 3, line 40): "The temperature measurements are done by analyzing Doppler broadened sideband spectra acquired by sideband spectroscopy (see e.g. [35, 36]). Further details can be found in Section Methods. T_{ref} was found ... "

New Section (Page 5): New section Method with text: "Each measurement of the axial ion temperature T_z is performed by analyzing the Doppler-broadened Gaussian envelope of a motional sideband spectrum. For this, sidebands are probed until excitation cannot be observed anymore. It is important to note that in order to reduce total measurement time, and therefore related errors arising from e.g. laser beam pointing fluctuations, only every second sideband is probed. This can be accomplished because the sideband transitions are separated in frequency by integer multiples of the axial frequency. In total, for each spectrum around 20 sidebands are acquired, where the number of sideband transitions depends on the Doppler broadening and thus the ion's axial temperature. For comparison, Figure 4 shows the resolved sideband spectrum measured after transport with the maximum and minimum heating observed in Fig. 3. The relative shift of the Gaussian fit center frequency in Fig. 4a and Fig. 4b results from slow magnetic field drifts, since both measurements were performed several weeks apart. In order to calculate the axial ion temperature T_z , the relation $T_z = m\lambda^2 v_D^2 / (8 \ln 2 k_B)$ is used [36], where m is the mass of the ${}^9\text{Be}^+$ ion, $\lambda \approx 313.165$ nm is the transition wavelength ${}^2\text{S}_{1/2} \leftrightarrow {}^2\text{P}_{3/2}$ (see Fig. 1 b) and v_D is the full width at half maximum of the fitted Gaussian envelope. For Fig. 4a and Fig. 4b, the values of v_D are determined to be (12.3 ± 0.6) MHz and (9.1 ± 0.5) MHz, respectively, from which the corresponding ion temperatures T_z of (2.88 ± 0.19) mK and (1.59 ± 0.18) mK are calculated. Together with T_{ref} , the heating resulting from each transport is calculated and shown in Fig. 3."

New Figure (Page 5): New Figure 4 with caption: "a, Resolved sideband spectrum after transport for a transport time of $t_{\text{tp}} = 7$ ms without filter predistortion. b, Resolved sideband spectrum after transport for a transport time of $t_{\text{tp}} = 15$ ms using filter predistortion. Each data point (blue) and the respective error (clear blue) are obtained from 50 Raman interrogations after transport. Red lines are Lorentzian fits to the sideband data, clear blue dashed lines are Gaussian envelope fits to the sideband maxima. $\nu_{\text{Ref}} = 138.909014$ GHz is a reference frequency close to ν_0 . Further details in the text."

2. Additionally, they state, "Probing only every second sideband comes with the advantage of a reduced total measurement time." Does this mean that after obtaining the entire spectrum with 20 transitions, they only scanned every second sideband in subsequent measurements to determine temperatures for each data point?

Response: A complete sideband spectrum comprises approximately 40 individual sidebands. To reduce the measurement time for each full spectral profile, approximately 20 individual sidebands were acquired instead of the typical 40. This was achieved by the acquisition of every second sideband. This is feasible due to the fixed frequency spacing between sidebands, which is determined by the axial frequency. We have introduced a method section, where we further discuss the temperature measurements and also introduced changes in the text. For both, please see our previous response.

3. Regarding the sentence after Eq. 1, I believe there should be no white space before "which is the input..." as it should be part of the same paragraph as the displayed equation.

Response: We have implemented the change in the text.

Reviewer #3

The authors present a novel technical achievement in the context of a precision measurement experiments, where a Be⁺ ion is transported within two separate stages of a multi-Penning trap stack. This is necessary as, in this and similar experiments, ions of different species are produced and held in different sections of the trap, to be then brought together to perform the measurement.

This work improves on results already presented by the same group, described in ref. [39]. The novelty in the current paper stands in the author's achievement of faster ion transport, still within the adiabatic regime, due to the inclusion of filter precompensation. The technique itself is not new, as it has been already demonstrated in the context of ion transport experiment focused on quantum computing (see e.g. ref [18] or <https://doi.org/10.1103/PhysRevA.102.022611>), where the transport distances and timescales are quite different. Nonetheless, their contribution is original and of high interest in their community, as (at the best of my knowledge) this is the first time such techniques are applied to trapped-ion-based precision measurement experiments, with demonstrated favorable results.

Their methodology is valid and clear, based on known techniques for the measurement of motional excitations in trapped ions, and their results are convincing. Overall, I would recommend the paper for publication. However, before the final publication, I suggest improving on a few points that in my opinion would help to increase the quality of the manuscript.

We are grateful for the reviewer taking the time to carefully review our manuscript and provide valuable feedback which has helped us to further improve it. We hope that the revised manuscript will be suitable. Below are our responses to each of the reviewer's comments.

Comments on the manuscript follow.

1. I think the paper lacks details on the temperature measurements. It is true that the technique is referenced elsewhere, e.g. in ref. [35], but it would be beneficial for this publication to give more in-context details, especially in view of directing the paper to a non-specialized audience. For example, the authors could show the data of one Doppler-broadened sideband spectrum as a representative measurement.

Response: We have introduced a new methods section in order to show further details on the temperature measurements. In addition, for comparison we have introduced a new figure 4. Here we show two sideband spectra after transport, one with large heating (7 ms without filter predistortion) and one without heating (15 ms with filter predistortion). We have also modified the manuscript accordingly.

Previous text (Page 3, line 27): "... sideband spectroscopy. For this, the carrier transition as well as every second sideband are probed until no excitation can be observed anymore. Here, around 20 transitions (including the carrier transition) are scanned. Probing only every second sideband comes with the advantage of a reduced total measurement time. Laser beam pointing fluctuations of a few tens of μm within an hour are observed, resulting in fluctuations of the observed excitation probability due to a decreased overlap of both Raman beams and the ion. T_{ref} was found to be ..."

New text (Page , line): "... sideband spectroscopy (see e.g. [35, 36]). Further details can be found in Section Methods. T_{ref} was found ... "

New Section (Page 5): New section Method with text: “Each measurement of the axial ion temperature T_z is performed by analyzing the Doppler-broadened Gaussian envelope of a motional sideband spectrum. For this, sidebands are probed until excitation cannot be observed anymore. It is important to note that in order to reduce total measurement time, and therefore related errors arising from e.g. laser beam pointing fluctuations, only every second sideband is probed. This can be accomplished because the sideband transitions are separated in frequency by integer multiples of the axial frequency. In total, for each spectrum around 20 sidebands are acquired, where the number of sideband transitions depends on the Doppler broadening and thus the ion's axial temperature. For comparison, Figure 4 shows the resolved sideband spectrum measured after transport with the maximum and minimum heating observed in Fig. 3. The relative shift of the Gaussian fit center frequency in Fig. 4a and Fig. 4b results from slow magnetic field drifts, since both measurements were performed several weeks apart. In order to calculate the axial ion temperature T_z , the relation $T_z = m\lambda^2 v_D^2 / (8 \ln 2 k_B)$ is used [36], where m is the mass of the ${}^9\text{Be}^+$ ion, $\lambda \approx 313.165$ nm is the transition wavelength ${}^2\text{S}_{1/2} \leftrightarrow {}^2\text{P}_{3/2}$ (see Fig. 1 b) and v_D is the full width at half maximum of the fitted Gaussian envelope. For Fig. 4a and Fig. 4b, the values of v_D are determined to be (12.3 ± 0.6) MHz and (9.1 ± 0.5) MHz, respectively, from which the corresponding ion temperatures T_z of (2.88 ± 0.19) mK and (1.59 ± 0.18) mK are calculated. Together with T_{ref} , the heating resulting from each transport is calculated and shown in Fig. 3.”

New Figure (Page 5): New Figure 4 with caption: “a, Resolved sideband spectrum after transport for a transport time of $t_{\text{tp}} = 7$ ms without filter predistortion. b, Resolved sideband spectrum after transport for a transport time of $t_{\text{tp}} = 15$ ms using filter predistortion. Each data point (blue) and the respective error (clear blue) are obtained from 50 Raman interrogations after transport. Red lines are Lorentzian fits to the sideband data, clear blue dashed lines are Gaussian envelope fits to the sideband maxima. $\nu_{\text{Ref}} = 138.909014$ GHz is a reference frequency close to ν_0 . Further details in the text.”

2. *Can the authors comment on the drastic increase in heating seen for $t_{\text{tp}} \leq 6$ ms, regardless of precompensation? Can they relate this timescale with other waveform parameters, e.g. the filter cutoff frequency?*

Response: The main reason is the strong distortion of the waveforms for shorter transport times due to the filter cut-off frequency. For 15-ms transport time and below, the waveforms start to get distorted due to the filters. Since the shorter the transport time, the larger the distortion of the waveform after passing the filters, an exponential increase of the heating for shorter transport times can be observed (See Figure 2a). It appears that $t_{\text{tp}} = 6$ ms represents an experimental threshold beyond which adiabatic transport is no longer feasible in our system. We have introduced the following text to the manuscript.

Previous text (Page 4, line 26): “... quanta (not shown in Fig. 3). These results demonstrate ...”

New text (Page 4, line 35): “... quanta (not shown in Fig. 3). The rapid increase in heating for $t_{\text{tp}} \leq 6$ ms can be attributed to severe distortion of the waveforms for shorter transport times due to the filter cut-off frequency, such that the distortion is too large to be fully compensated. Nevertheless, these results demonstrate ...”

3. *In Fig.2, it would be clearer to plot on the same axes the voltage waveforms with and without filter precompensation, rather than plotting the difference between the two on a separate axis. Furthermore, it would be useful to add a color-coded sketch of the trap electrodes, perhaps similar to the one in ref. [39],*

fig 1, matching the electrode colors with the corresponding waveform plot – or with a similar visual reference.

Response: In Figure 1a, we have introduced a color coding in the trapped electrodes according to their respective transport voltage waveform shown in Figure 2. In addition, we have introduced the voltage waveforms without filter predistortion in Figure 2a as dashed lines. We have also modified the text according to the changes.

Previous text (Page 2, line 22, Caption Figure 1): "... noise. b, Relevant ..."

New text (Page 2, line 22, Caption Figure 1): "... noise. The trap electrodes are color-coded according to their respective transport voltage waveform shown in Figure 2. b, Relevant ..."

Modified Figure (Page 2): Modification in Figure 1a.

Previous text (Page 3, line 27, Caption Figure 2): "Figure 2. a, Voltage waveforms with filter predistortion as a function of ..."

New text (Page 3, line 27, Caption Figure 2): "Figure 2. a, Voltage waveforms with (solid lines) and without (dashed lines) filter predistortion as a function of ..."

Modified Figure (Page 3): Modification in Figure 2a.

Other corrections to the manuscript

During the revision of the paper, we found a few minor errors. These have been corrected in the manuscript.

Previous text (Page 3, line 19): "R2 and R3 were estimated from measured resistance values for several temperatures close to 4 K.

New text (Page 3, line 33): "R2 and R3 were estimated from measured resistance values for several temperatures close to 60 K and 5 K, respectively."

Previous text (Page 4, line 20): "... no rapid increase of the induced heating can be observed and an average heating of 2.6 ± 4.0 quanta is obtained for transport times $7 \text{ ms} \leq t_{\text{tp}} \leq 9 \text{ ms}$."

Previous text (Page 4, line 29): "... no rapid increase of the induced heating can be observed and an average heating of 2.6 ± 4.0 quanta is obtained for transport times $7 \text{ ms} \leq t_{\text{tp}} \leq 15 \text{ ms}$."

Dear editors of Communications Physics,

We would like to thank you again for your valuable time spent reviewing our manuscript. We have introduced a few changes to the manuscript following the style guidelines and the editorial checklist of communications physics. With best regards from the authors,

Moritz von Boehn,
Juan M. Cornejo and
Christian Ospelkaus

Reviewer #1

The authors have addressed all my questions and comments.

Reviewer #2

The authors have addressed all of my concerns in their responses, and the new version of the manuscript is clarified. Therefore, I recommend the manuscript for publication.

Reviewer #3

I thank the authors for the time taken and the effort put in replying to all referees' comments. I think that this clarified all questioned points and definitely improved the manuscript. I recommend their work for publication.

We are grateful for the reviewers taking the time to carefully review our manuscript and provide valuable feedback which has helped us to further improve it.

Corrections to the manuscript

- Subscripts and superscripts have been set in Roman when they are labels (including figures).
- Unit dimensions have been expressed using the word "per".
- "Discussion" heading has been removed and "Results" heading has been changed to "Results and Discussion", as the Discussion section is not an extended analysis.
- We have specified in the captions of Figures 3 and 4 what the displayed error bars represent.
- Titles in bold have been added in each caption of the Figures.
- As reference citations will be superscripted in the final version of the manuscript, in citations as 'see Ref. ¹' and 'see e.g. Ref. ¹' we removed 'see Ref.' and 'see e.g. Ref.' in the revised manuscript. Instead we directly give the reference. In addition, instead of "...can be found in Ref. ¹.' we now use "...can be found elsewhere¹'.